# Comparison of the *Opn-CreER* and *Ck19-CreER* Drivers in Bile Ducts of Normal and Injured Mouse Livers

**DOI:** 10.3390/cells8040380

**Published:** 2019-04-25

**Authors:** Bram Lesaffer, Elisabeth Verboven, Leen Van Huffel, Iván M. Moya, Leo A. van Grunsven, Isabelle A. Leclercq, Frédéric P. Lemaigre, Georg Halder

**Affiliations:** 1VIB Center for Cancer Biology and KU Leuven Department of Oncology, University of Leuven, 3000 Leuven, Belgium; bram.lesaffer@student.kuleuven.be (B.L.); elisabeth.verboven@kuleuven.vib.be (E.V.); leen.vanhuffel@kuleuven.vib.be (L.V.H.); ivan.moya@kuleuven.vib.be (I.M.M.); 2Facultad de Ingeniería y Ciencias Aplicadas, Universidad de Las Americas, 170501 Quito, Ecuador; 3Liver Cell Biology research group, Vrije Universiteit Brussel, 1090 Brussels, Belgium; leo.van.grunsven@vub.be; 4Laboratory of Hepato-gastroenterology, Institut de Recherche Expérimentale et Clinique, Université catholique de Louvain, 1200 Brussels, Belgium; isabelle.leclercq@uclouvain.be; 5Liver and Pancreas Development Unit, de Duve Institute, Université catholique de Louvain, 1200 Brussels, Belgium; frederic.lemaigre@uclouvain.be

**Keywords:** Cre, cholangiocytes, bile duct cells, knockout, mouse liver, lineage tracing, Opn, Ck19

## Abstract

Inducible cyclization recombinase (Cre) transgenic mouse strains are powerful tools for cell lineage tracing and tissue-specific knockout experiments. However, low efficiency or leaky expression can be important pitfalls. Here, we compared the efficiency and specificity of two commonly used cholangiocyte-specific Cre drivers, the *Opn-iCreER^T2^* and *Ck19-CreER^T^* drivers, using a tdTomato reporter strain. We found that *Opn-iCreER^T2^* triggered recombination of the tdTomato reporter in 99.9% of all cholangiocytes while *Ck19-CreER^T^* only had 32% recombination efficiency after tamoxifen injection. In the absence of tamoxifen, recombination was also induced in 2% of cholangiocytes for the *Opn-iCreER^T2^* driver and in 13% for the *Ck19-CreER^T^* driver. For both drivers, Cre recombination was highly specific for cholangiocytes since recombination was rare in other liver cell types. Toxic liver injury ectopically activated *Opn-iCreER^T2^* but not *Ck19-CreER^T^* expression in hepatocytes. However, ectopic recombination in hepatocytes could be avoided by applying a three-day long wash-out period between tamoxifen treatment and toxin injection. Therefore, the *Opn-iCreER^T2^* driver is best suited for the generation of mutant bile ducts, while the *Ck19-CreER^T^* driver has near absolute specificity for bile duct cells and is therefore favorable for lineage tracing experiments.

## 1. Introduction

The development of inducible Cre transgenic mouse strains has made it possible to generate cell type-specific knockout and lineage tracing experiments. Such experiments generally require two genetically modified loci. The first locus contains a gene or an artificial transgene that is flanked by two *loxP* recombination sites, a so-called “floxed” allele. The second locus is a transgene or a knock-in where a cell type-specific promoter drives the expression of the Cre recombinase. Often this transgene encodes a tamoxifen-inducible Cre where Cre is fused to the ligand binding domain of the Estrogen receptor (CreER). In the absence of tamoxifen, CreER is sequestered in the cytoplasm. Upon tamoxifen binding, CreER is released from the cell membrane and translocates into the nucleus where it catalyzes recombination between the two *loxP* sites. This results in the excision of a DNA segment that is flanked by two equally oriented *loxP* sites. This inducible CreER-loxP system is broadly used to conditionally knock out a gene of interest in specific cells by combining tissue specific CreER strains with tamoxifen injection at a specific time point. This conditional knock-out strategy thus prevents deletion of floxed segments prematurely or in non-desired cell types.

The CreER-loxP system can also be used for lineage tracing. Here, CreER activation is used to excise a DNA segment that contains a floxed transcriptional *STOP* cassette (*loxP-STOP-loxP* cassette) that separates a promoter from a reporter gene. Often used are strains with a *loxP-STOP-loxP* cassette followed by the coding region of a fluorescent protein inserted into the ubiquitously expressed *Rosa26* locus. Upon tamoxifen injection, CreER deletes the *STOP* cassette and causes expression of the reporter gene in the targeted cells and their progeny.

A large number of mouse strains have been generated that express different versions of Cre in specific cell types. However, Cre lines often do not show 100% efficiency and specificity, two factors that can greatly affect the outcome of a gene knockout experiment. On the one hand, inefficient gene deletion produces mosaic tissues where non-recombined wild-type cells may compensate for the phenotype of mutant cells. On the other hand, recombination in non-target cells caused by unspecific Cre expression may produce misleading phenotypes. Similarly, in lineage tracing experiments, low labeling efficiency can cause underestimation of the contribution of the traced cell type and unspecific reporter activation makes it impossible to know the origin of the labeled cells. Thus, a thorough analysis of the efficiency and specificity of a Cre driver is required to correctly execute knockout and lineage tracing experiments.

A number of different Cre strains have been generated to trigger *loxP* recombination in cholangiocytes, which are the cells that constitute the bile ducts and play important and specialized roles in development, homeostasis, and regeneration of the liver [1]. These drivers include *Ck19-CreER^T^* [2], *Hnf1b-CreER* [3], *Sox9-IRES-CreER^T2^* [4], and *Opn-iCreER^T2^* [5]. Although all of these transgenes drive *CreER* in adult cholangiocytes, they display notable differences in their efficiency and specificity which can result in data misinterpretation and lack of reproducibility.

The *Hnf1b-CreER* driver has 84% recombination efficiency and is highly specific for cholangiocytes, since only 0.0125% of hepatocytes had recombination of a *YFP* reporter transgene [6,7]. However, use of the *Hnf1b-CreER* driver in cholangiocyte research has been limited, likely because *Hnf1b* is expressed in hepatocytes during the early postnatal period [8]. The *Sox9-IRES-CreER^T2^* driver was initially used for lineage tracing of bile duct cells upon liver injury [4], but broad ectopic expression of this driver (and also the *Sox9* gene) in hepatocytes upon liver injury resulted in confounding cell tracing results [6,9,10,11]. However, a different *Sox9-CreER* line generated later showed higher specificity for bile duct cells when used with low doses of tamoxifen [6,9,10,11]. The *Ck19-CreER^T^* driver is frequently used and regarded as highly specific towards cholangiocytes; however, this driver suffers from low efficiency in deleting floxed alleles [2,5]. Curiously, another *Ck19-Cre* driver triggered recombination in cholangiocytes and a subset of hepatocytes [12]. Finally, *Opn-iCreER^T2^* showed high recombination efficiency in cholangiocytes [5]. However, the *Opn* gene is ectopically expressed in hepatocytes and other liver cells upon chronic injury [13,14,15], raising questions about the specificity of the *Opn-iCreER^T2^* driver.

In this study, we compared the efficiency and specificity of *Cre* mediated recombination triggered by the *Ck19-CreER^T^* and *Opn-iCreER^T2^* drivers. Our results show that both Cre drivers nearly exclusively target cholangiocytes but that recombination in hepatocytes can occasionally occur in either model after tamoxifen injection, although this is a rare event. However, the *Opn-iCreER^T2^* driver showed vastly superior recombination efficiency.

## 2. Materials and Methods

### 2.1. Mouse Strains

Opn-iCreER^T2^-Rosa26-loxP-STOP-loxP-tdTomato mice (Opn-CreER-tdTomato) were generated by crossing Opn-iCreER^T2^ mice with Rosa26-loxP-STOP-loxP-tdTomato (R26-tdTomato) mice. Opn-iCreER^T2^ mice were previously described [5]. R26-tdTomato mice were kindly provided by Chris Marine [16]. Ck19-CreER^T^-Rosa26- loxP-STOP-loxP-tdTomato mice (Ck19-CreER-tdTomato mice) were generated by crossing Ck19-CreER^T^ mice with R26-tdTomato mice. Ck19-CreER^T^ mice were purchased from The Jackson Laboratory (JAX stock #026925; Bar Harbor, Maine) [2]. Male and female mice were used in all experiments. Mice were housed, fed, and treated in accordance with protocols approved by the committee for animal research at KU Leuven. All mouse experiments were approved by the institutional ethical commission at KU Leuven (P146-2017) and were performed in accordance with relevant institutional and national guidelines and regulations.

### 2.2. Tamoxifen and CCl_4_ Injections

Tamoxifen (T5648; Sigma, St. Louis, MI, USA) was dissolved in corn oil at a concentration of 10 mg/mL and injected intraperitoneally for five consecutive days at a dose of 80 mg/kg bodyweight into mice aged 6 to 8 weeks. Mice from uninjured groups were sacrificed after either a 3-day or a 3-week washout period following the last tamoxifen injection. Injured groups received an intraperitoneal injection of CCl_4_ (1 mL/kg bodyweight, in corn oil) after a 3-day or a 3-week washout period. Mice were sacrificed 3 days after CCl_4_ injection. Mice with liver injury but without tamoxifen washout received CCl_4_ on day 2 of the 5 days of tamoxifen injection. These mice were sacrificed on day 5, the last day of tamoxifen injection. Vehicle control mice were injected with corn oil on five consecutive days, mice were sacrificed 3 days after corn oil injection.

### 2.3. Immunostaining and Imaging

Livers were collected and fixed in 4% paraformaldehyde (PFA) in PBS for 48 h at 4 °C and then washed in PBS. Liver lobes were embedded in 4% agarose in PBS and sectioned at 100 μm thickness using a vibratome (model VT 1000S; Leica, Wetzlar, Germany). Liver sections were permeabilized and antigens were retrieved with TRIS-EDTA for 20 min at 95 °C and blocked in 3% Bovine Serum Albumin (BSA) in PBS for 2 h at room temperature. The sections were then incubated in primary antibody solution overnight. The following day, sections were washed and then incubated in secondary antibody solution for 1 h at room temperature. Finally, sections were washed and mounted in mowiol, and analyzed on an FV1200 confocal microscope (Olympus, Tokio, Japan). Images were processed in ImageJ (Version 2.0.0-RC-69/1.52n, open source). Primary antibodies were rat-anti-Ck19 (TROMA III, 1:50; Hybridoma Bank, Iowa City, IA, USA), rabbit-anti-Hnf-4α (ab181604, 1:200; Abcam, Cambridge, UK), goat-anti-Opn (AF143, 1:250; R&D systems, Minneapolis, MN, USA), rabbit-anti-Cd45 (Abcam, ab10558, 1:200) and rabbit-anti-Desmin (RB-9014-P, 1:200; Thermo Fischer Scientific, Waltham, MA, USA). Secondary antibodies were donkey anti-rat-488 (code: 712-225-153, 1:500; Jackson, Bar Harbor, ME, USA), donkey anti-rabbit-647 (A31573, 1:500; Invitrogen, Carlsbad, CA, USA), and donkey anti-goat 488 (Jackson, code: 705-545-147, 1:500) and donkey anti-rabbit-488 (Jackson, code: 711-545-152). DAPI (1:1000; Thermo Fisher Scientific, Waltham, MA, USA) was used to stain nuclei.

### 2.4. Quantitative Analysis

All quantifications were performed using the Cell Counter plugin in ImageJ software. For both efficiency and specificity experiments, sections from the same mice were analyzed. Heterozygous *Opn-CreER-tdTomato* mice (*n* = 30 mice) and heterozygous *Ck19-CreER-tdTomato* mice (*n* = 31 mice) were distributed across different treatment groups. *Opn-CreER-tdTomato* groups were uninjured 3-day after Tamoxifen (*n* = 7), uninjured 3-week after Tamoxifen (*n* = 8), CCl_4_-treated 3-day washout (*n* = 4), CCl_4_-treated 3-week washout (*n* = 3), CCl_4_-treated no washout (*n* = 5), and corn oil controls (*n* = 3). *Ck19-CreER-tdTomato* groups were uninjured 3-day after Tamoxifen (*n* = 7), uninjured 3-week after Tamoxifen (*n* = 8), CCl_4_-treated 3-day washout (*n* = 5), CCl_4_-treated 3-week washout (*n* = 4), CCl_4_-treated no washout (*n* = 5), and corn oil controls (*n* = 2). Homozygous *Opn-CreER-tdTomato* mice (*n* = 3) and homozygous *Ck19-CreER-tdTomato* mice (*n* = 4) were checked for bile duct phenotypes.

Hnf-4α^+^ hepatocytes were quantified by counting 18 pictures from two liver sections per mouse. This amounted to about 5000 Hnf-4α^+^ cells per section (4697 ± 704). The number of Hnf-4α^+^,tdTomato^+^ double positive cells was divided by the total number of Hnf-4α^+^ cells for each mouse. The resulting fraction was expressed as percentage and used as a measure for *R26-tdTomato* recombination.

To determine the efficiency of *R26-tdTomato* recombination in cholangiocytes, per mouse 20 bile duct regions of all sizes were visualized in one liver section and Ck19^+^ cells were counted. On average 400 (397 ± 69) Ck19^+^ cells were counted per mouse. Every Ck19^+^ cell was checked for tdTomato expression. The number of Ck19^+^,tdTomato^+^ double positive cells was then divided by the total number of Ck19^+^ cells, expressed as a percentage, and used as a measure for recombination efficiency.

### 2.5. Statistical Analysis

Two-way ANOVA with interaction was performed on percentages of Hnf-4α^+^ cells that were also tdTomato^+^ to test for differences in specificity between genotypes and between washout types. In the two-way ANOVA for uninjured mice genotypes included *Opn-CreER-tdTomato and Ck19-CreER-tdTomato* and washout groups included 3-day washout or 3-week washout. In the two-way ANOVA for CCl_4_-injured mice genotypes included *Opn-CreER-tdTomato and Ck19-CreER-tdTomato* and washout types included 3-day washout, 3-week washout, or no washout. Tukey’s range test was performed as a post-hoc test to compare all possible pairs of means in both ANOVAs. Two-way ANOVA was performed on percentages of Ck19^+^ cells that were also tdTomato^+^ to test for differences in efficiency between genotypes and treatment groups. Genotypes included *Opn-CreER-tdTomato* and *Ck19-CreER-tdTomato*, and treatment groups included uninjured or CCl_4_-treated after either a 3-day or 3-week washout following tamoxifen injections, resulting in a total of 8 groups. Tukey’s range test was performed as a post-hoc test to compare all possible pairs of means. A *p*-value of 0.05 and lower was considered to be statistically significant for all tests. All statistical tests were performed using R-Studio (Version 1.1.463; R studio Inc., Boston, MA, USA).

## 3. Results

### 3.1. Opn-iCreER^T2^ Drives loxP Site Recombination more Efficiently than Ck19-CreER^T^

In order to evaluate the efficiency and specificity of Cre-mediated recombination in mice expressing CreER recombinase under the control of the *Cytokeratin 19 (Ck19)* and *Osteopontin (Opn)* promoters we crossed the *Ck19-CreER^T^* and *Opn-iCreER^T2^* mice with mice containing the Cre-recombination reporter *Rosa26*-*loxP-STOP-loxP-tdTomato (R26-tdTomato)*. This generated mice with the *Ck19-CreER^T^* or *Opn-iCreER^T2^* and the *R26-tdTomato* transgenes, herewith referred to as *Ck19-CreER-tdTomato* and *Opn-CreER-tdTomato* mice respectively (Figure 1a).

First, we evaluated whether the two Cre drivers induced recombination in the absence of tamoxifen. Liver sections were immunostained for Ck19 to mark all cholangiocytes, and were imaged for Ck19 and tdTomato expression. After corn oil injection (vehicle), *Opn-CreER-tdTomato* mice showed tdTomato expression in 2% of cholangiocytes (9/388). The 9 tdTomato^+^ cholangiocytes were found in 7 different bile ducts and mostly appeared as single cells. In *Ck19-CreER-tdTomato* corn oil injected mice, 13% of cholangiocytes were tdTomato^+^ (54/418) (Figure 1b). The 54 tdTomato^+^ cells were found in 8 different bile ducts and appeared as large patches of cells, presumably clones, in large bile ducts. This amounted to 1.80 independent recombination events (clones) per 100 cholangiocytes for *Opn-CreER^T^-tdTomato* mice and 1.91 for *Ck19-CreER^T^-tdTomato* mice observed at 8 weeks of age. We did not observe tdTomato expression in any other liver cell type in either strain. Thus, while both strains showed some degree of leakiness, *Ck19-CreER^T^* had high numbers of recombined cells in the uninduced condition.

Next, we evaluated the recombination efficiency induced by tamoxifen administration. We injected five doses of tamoxifen on consecutive days and analyzed tdTomato expression three days after the last injection. In *Ck19-CreER-tdTomato* mice, tdTomato expression was highly mosaic and we did not detect any duct that was entirely composed of tdTomato^+^ cells (Figure 1b, Supplemental Appendix A). Rather, the percentage of tdTomato^+^ cholangiocytes per bile duct was variable, and in most ducts less than half of the Ck19^+^ cells expressed tdTomato (Figure 1b,c). Quantification of a large number of bile ducts showed that only 35% of the Ck19^+^ cholangiocytes (932/2676) expressed tdTomato (Figure 1d,e; Table 1). This number did not increase over time and three weeks after tamoxifen injection still only about 28% of the Ck19^+^ cells expressed tdTomato (862/3124). In contrast, virtually all bile duct cells expressed tdTomato in *Opn-CreER-tdTomato* mice (Figure 1b). Quantification showed that tdTomato was detected in 99.96% and 99.94% of the cholangiocytes (2722/2723; 3298/3300) three days and three weeks after tamoxifen injection, respectively (Figure 1d,e; Table 1). These data show that the *Opn-iCreER^T2^* driver efficiently recombines the floxed *STOP* cassette whereas the *Ck19-CreER^T^* driver has limited recombination efficiency when using *R26-tdTomato* as a reporter.

During this analysis, we also observed that heterozygous *Ck19-CreER-tdTomato* mice had reduced Ck19 expression. This is probably due to the fact that the *Ck19-CreER^T^* construct is a knock-in into the Ck19 locus causing loss of function (Appendix A) [2]. Homozygous *Ck19-CreER-tdTomato* mice lacked Ck19 expression altogether (Appendix A). On the other hand, neither Ck19 nor Opn expression was affected in heterozygous or homozygous *Opn-CreER-tdTomato* mice consistent with the transgenic nature of the *Opn-iCreER^T2^* construct (Appendix A) [5].

### 3.2. The Ck19-CreERT and Opn-iCreERT2 Drivers are Highly Specific towards Cholangiocytes

To determine the frequency of recombination in hepatocytes, we stained the *Ck19-CreER-tdTomato* and *Opn-CreER-tdTomato* mice for the hepatocyte-specific marker Hnf-4α and determined overlap with tdTomato expression. No tdTomato^+^ hepatocytes were detected in *Ck19-CreER-tdTomato* mice three days (0/61190) or three weeks (0/76513) after the last tamoxifen injection (Figure 2; Table 2). *Opn-CreER-tdTomato* mice had very few tdTomato^+^ hepatocytes, namely 5 hepatocytes out of 67462 (0.0074%) three days after tamoxifen injection and 7 hepatocytes out of 77086 (0.0091%) three weeks after tamoxifen treatment (Figure 2; Table 2). These tdTomato^+^ hepatocytes usually appeared as single cells and their location was not restricted to a specific zone. All tdTomato^+^ Ck19^-^ cells were positive for Hnf-4α, and none of those cells co-stained for the hepatic stellate cell marker Desmin or the immune cell marker Cd45 (Appendix A), confirming earlier reports [17]. We conclude that the *Ck19-CreER^T^* driver had absolute specificity in these experiments while the *Opn-iCreER^T2^* driver had extremely low frequency of recombination in cells other than cholangiocytes.

### 3.3. Ectopic Expression of the Cre Drivers after Liver Injury

Liver injury can affect gene expression in diverse liver cells. As such, ectopic Opn expression was observed in hepatocytes, HSCs, and immune cells after liver injury [13,15,18,19]. We therefore assessed the specificity of the *Ck19-CreER^T^* and the *Opn-iCreER^T2^* drivers after liver injury caused by carbon tetrachloride (CCl_4_) toxicity. To do this, we injected a single dose of CCl_4_ into mice either during the tamoxifen treatment, or three days or three weeks after the end of the tamoxifen treatment. We then quantified the number of Hnf-4α^+^ cells that expressed tdTomato three days after the CCl_4_ injection (Figure 3a–c). As shown by H&E staining, CCl_4_ induced necrosis in pericentral hepatocytes (Appendix A). In the *Ck19-CreER-tdTomato* mice, we found only one of over 100,000 hepatocytes that expressed tdTomato (1/21016; 0/44034; 0/40768 tdTomato^+^ hepatocytes for the three conditions). This result indicates that CCl_4_ does not induce ectopic expression of the *Ck19-CreER^T^* transgene (Figure 3a–c, Table 2). Indeed, Ck19 staining of CCl_4_-injected mice showed that Ck19 expression remained restricted to cholangiocytes after liver injury (Figure 3d). Similarly, extremely few tdTomato^+^ hepatocytes (0/35801; 7/30984) were found in *Opn-CreER-tdTomato* mice when CCl_4_ was injected three days or three weeks after tamoxifen administration (Figure 3a,b). However, when CCl_4_ was injected during tamoxifen treatment, 0.33% of hepatocytes (81/24192) in the *Opn-CreER-tdTomato* mice expressed tdTomato (Figure 3c–d; Table 2). Thus, elevated ectopic recombination in hepatocytes was observed when the injury was concomitant with the tamoxifen administration. This is consistent with the ectopic expression of the endogenous *Opn* gene after different types of liver injury [13,19]. No tdTomato expressing HSCs or immune cells were detected in either mouse model, as shown by immunostaining for Desmin and Cd45 respectively (Appendix A). Recombination efficiencies in cholangiocytes were comparable between injured and non-injured conditions (Table 1). Altogether, these data show that *Ck19-CreER^T^* and *Opn-iCreER^T2^* retain their high specificity towards cholangiocytes after liver injury. However, CCl_4_ induces ectopic Opn expression in hepatocytes and therefore a 3-day washout period between tamoxifen and toxin treatment is required to maintain this high level of specificity of the *Opn-iCreER^T2^* driver when using *R26-tdTomato* as a reporter.

## 4. Discussion

In this study we performed a thorough analysis of the efficiency and specificity by which the *Ck19-CreER^T^* and *Opn-iCreER^T2^* drivers trigger recombination of the *R26-tdTomato* reporter in the liver.

We followed a standardized tamoxifen regimen and compared tdTomato expression in homeostatic livers and livers with acute toxic injury. Our results show that the *Ck19-CreER^T^* driver has near absolute specificity for cholangiocytes under normal and injury conditions. However, *Ck19-CreER^T^* has relatively low efficiency such that only about 32% of cholangiocytes underwent recombination of the *R26-tdTomato* reporter. On the other hand, the *Opn-iCreER^T2^* driver showed nearly 100% efficiency in recombining the *R26-tdTomato* reporter in cholangiocytes. Notably, other studies reported lower levels of recombination with the *Opn-iCreER^T2^* driver, probably because lower amounts of tamoxifen were injected and a less sensitive reporter strain was used (*Rosa26-loxP-STOP-loxP-YFP*) [5,17]. In any case, our findings indicate that *Opn-iCreER^T2^* drives *LoxP* site recombination more efficiently than the *Ck19-CreER^T^* driver and is therefore more suitable to perform knockout studies in bile ducts.

Despite the fact that *Opn-iCreER^T2^* is a stronger recombination driver than *Ck19-CreER^T^*, one should keep in mind that efficiency of creating knockout cells may be lower than 100% in knockout studies. Recombination of one allele is sufficient to produce a positive signal from the *R26-tdTomato* reporter, yet generation of homozygous mutant cells may require recombination of both alleles. In addition, the sensitivity to Cre recombination for a floxed allele may be lower than for the tdTomato reporter line, which is known to be very sensitive [20]. Determination of the recombination efficiency is therefore important to interpret the results of knockout experiments.

Our study showed Cre activity in the absence of tamoxifen. Recombination occurred in 13% of cholangiocytes in the *Ck19-CreER^T^-tdTomato* line and in 2% in the *Opn-iCre-ER^T2^-tdTomato* line. Despite its high specificity, the *CK19-CreER^T^* might therefore be unsuited for lineage tracing experiments since temporal control over Cre recombination is partially lost. These baseline levels of Cre recombination have to be considered when performing experiments.

Some hepatocytes ectopically activated *Opn-iCreER^T2^* and expressed the tdTomato reporter. This number was minimal, since less than 1 in 10,000 hepatocytes (0.01%) was positive for tdTomato in all conditions. This extremely low level of Cre recombination in hepatocytes may not be sufficient to affect the phenotype of a full knockout in bile ducts. Nevertheless, this low background recombination may make the *Opn-iCreER^T2^* driver unsuited for lineage tracing experiments that relay on absolute specificity for cholangiocytes, yet the driver can be used to generate mutant bile ducts.

The origin of the occasionally observed tdTomato^+^ hepatocytes is unclear. One possibility is that these hepatocytes were derived from cholangiocytes by trans-differentiation. However, we think that this scenario is improbable because we did not detect signs of a transition from cholangiocytes to hepatocytes, like expression of Opn or Ck19 in tdTomato^+^ Hnf-4α^+^ hepatocytes, at any analyzed time point, and because the tdTomato^+^ hepatocytes were not accumulated around bile ducts as would be expected if they were derived from cholangiocytes. Rather, the tdTomato expression in sparse hepatocytes may be caused by ectopic activation of the promoter driving Cre in hepatocytes. Indeed, expression of *Opn* is ectopically activated in hepatocytes upon liver injury, indicating that also the *Opn-iCreER^T2^* construct may be ectopically activated [13,19]. However, non-specific recombination driven by the ectopic activation of CreER in hepatocytes could be avoided when liver injury was induced three or more days after the end of the tamoxifen administration. This washout period was thus sufficient to lower the levels of tamoxifen below the threshold required to activate the CreER^T2^ that was ectopically expressed in hepatocytes [21]. The length of such a washout period will however depend on the tamoxifen regimen and on the injury model applied, and should therefore be re-examined when using a different experimental set-up. In addition, high doses of tamoxifen can cause liver injury and induce the expression of cholangiocyte-specific markers such as Sox9 in hepatocytes [22]. Thus, tamoxifen toxicity may contribute to the ectopic recombination of tdTomato in hepatocytes.

*Ck19-CreER^T^* was generated by a knock-in of the *CreER^T^* ORF at the start codon of the *Ck19* gene. This produced a *Ck19* loss-of-function mutation and indeed we observed reduced amounts of Ck19 expression in *Ck19-CreER^T^* heterozygous mice. Homozygous *Ck19-CreER^T^* mice are viable and appear to have normal bile ducts, although without Ck19 expression [23]. The possibility that reduced expression of Ck19 might impact cholangiocyte function or specific post-injury phenotypes needs to be further evaluated. The *Opn-iCreER^T2^* mice were created by random integration of a bacterial artificial chromosome (BAC) containing the *Opn* gene with *iCreER^T2^* inserted. The insertion of this construct into the mouse genome did not cause detectable deleterious effects, as homozygous *Opn-iCreER^T2^* mice are viable and did not show altered bile duct morphology or cholangiocyte gene expression. Thus, if desired, both drivers can be bred to homozygosity.

## 5. Conclusions

Our data indicate that the *Opn-iCreER^T2^* driver is best suited for the generation of mutant bile ducts, while the *Ck19-CreER^T^* driver has near absolute specificity for bile duct cells and is therefore favorable for lineage tracing experiments. Importantly however, it has to be considered in this model that Cre recombination is present in the absence of tamoxifen.

## Figures and Tables

**Figure 1 cells-08-00380-f001:**
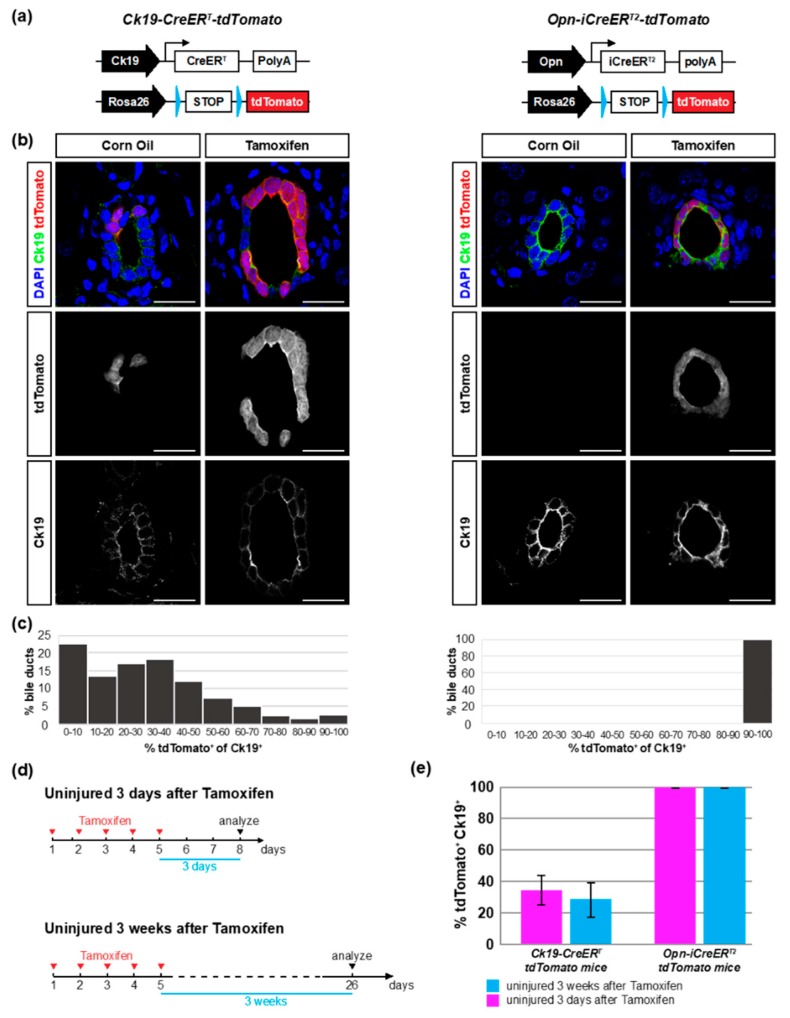
*Opn-iCreER^T2^* drives *loxP* site recombination more efficiently than *Ck19-CreER^T^*. (**a**) Schematic overview of Cre drivers and recombination alleles of *Ck19-CreER-tdTomato* and *Opn-CreER-tdTomato* mice. (**b**) Immunofluorescent detection of Ck19 and tdTomato in liver sections of *Ck19-CreER-tdTomato* and *Opn-CreER-tdTomato* mice after vehicle or tamoxifen injection. (**c**) Distribution of recombination efficiency in individual bile ducts ranging from 0 to 100 percent. (**d**) Schematic experimental outline. Mice were injected with tamoxifen on five consecutive days and sacrificed three days or three weeks later. (**e**) Percentage of tdTomato^+^ Ck19^+^ cells quantified in liver sections at different timepoints after tamoxifen injection (*n* = 7–8 mice per experiment; 20 bile duct regions per mouse). The difference between the two genotypes was significant 3 days (*p* = 0.001) and 3 weeks (*p* = 0.001) after tamoxifen injection. Scale bars: 20 μm.

**Figure 2 cells-08-00380-f002:**
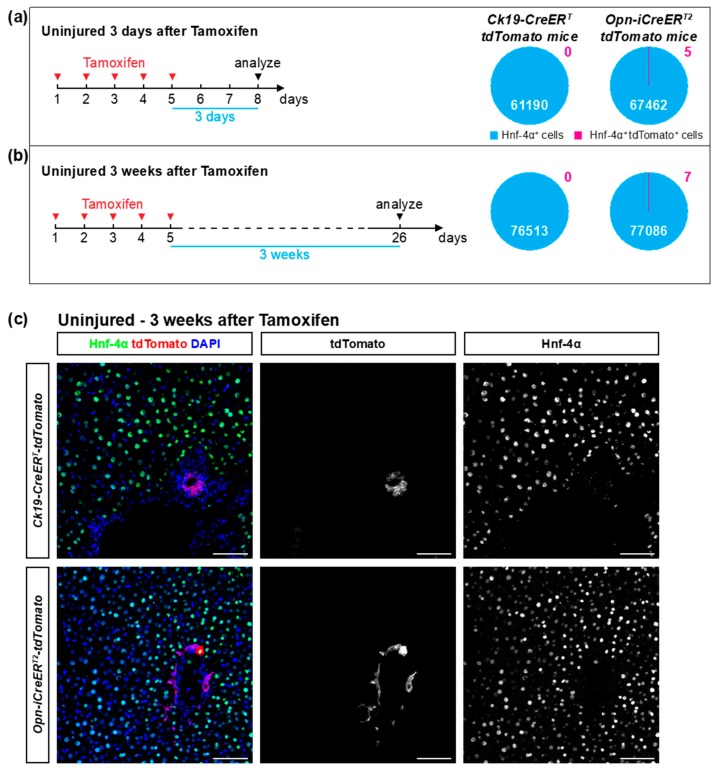
The *Ck19-CreER^T^* and *Opn-iCreER^T2^* drivers are highly specific towards cholangiocytes in non-injured livers. (**a**,**b**) Schematic experimental outline. Mice were injected with tamoxifen and sacrificed three days or three weeks later. The percentage of tdTomato^+^ Hnf-4α^+^ cells was quantified in liver sections of *Ck19-CreER-tdTomato* and *Opn-CreER-tdTomato* mice (*n* = 7–8 mice per experiment; 18 pictures per mouse). No statistically significant differences were detected between groups. (**c**) Immunofluorescent detection of Hnf-4α and tdTomato in *Ck19-CreER-tdTomato* and *Opn-CreER-tdTomato* mice three weeks after tamoxifen. Scale bars: 50 μm.

**Figure 3 cells-08-00380-f003:**
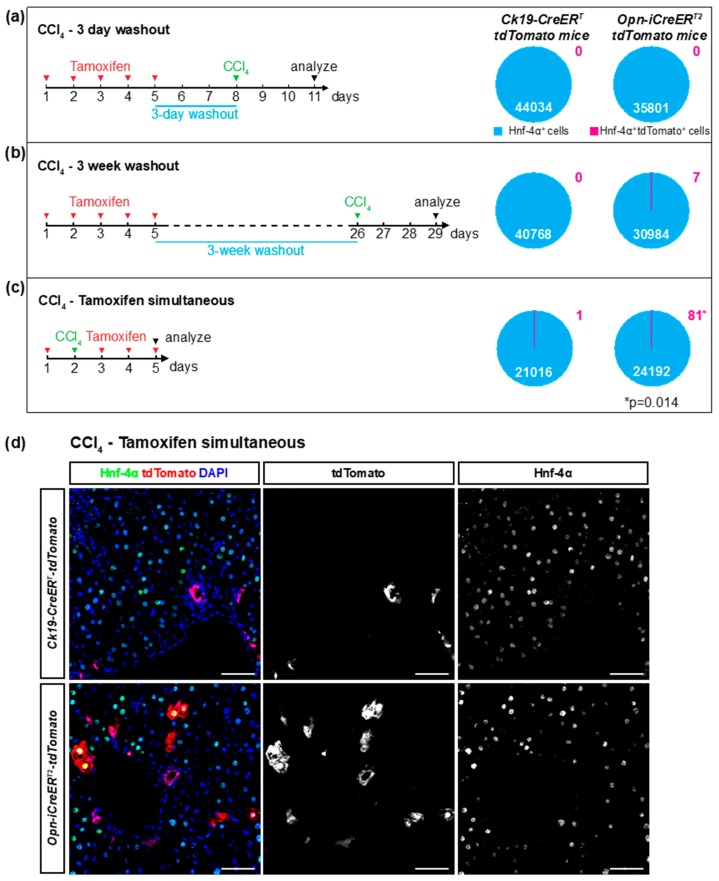
Ectopic activation of Cre drivers in hepatocytes after liver injury. (**a**–**c**) Schematic experimental outline. Mice were injected with CCl_4_ after a three-day washout (**a**), a three-week washout (**b**) or no washout (**c**) after tamoxifen and sacrificed three days later. The percentage of tdTomato^+^ Hnf-4α^+^ cells was quantified in liver sections of *Ck19-CreER-tdTomato* and *Opn-CreER-tdTomato* mice (*n* = 3–5; 18 pictures per mouse). (**d**) Immunofluorescent detection of Hnf-4α and tdTomato on liver sections of mice treated with tamoxifen and CCl_4_ without washout period. Scale bars: 50 μm.

**Table 1 cells-08-00380-t001:** Quantification of tdTomato+ cholangiocytes.

Treatment Group	Mice (*n*)		$\frac{\#\;\mathrm{Ck}19^{+}\;{\mathrm{tdTomato}}^{+}}{\#\;\mathrm{Ck}19^{+}}$		% tdTomato^+^ of Ck19^+^	
	Opn-Cre	Ck19-Cre	Opn-Cre	Ck19-Cre	Opn-Cre	Ck19-Cre
Uninjured 3 days after TAM	7	7	2722/2723	932/2676	99.9 ± 0.1	34.8 ± 9.1
Uninjured 3 weeks after TAM	8	8	3298/3300	862/3124	99.9 ± 0.1	27.6 ± 11.7
CCl_4_ 3-day washout	4	5	1621/1621	731/1880	100	38.9 ± 10.8
CCl_4_ 3-week washout	3	4	1326/1328	442/1588	99.9 ± 0.2	27.8 ± 5.3

Table showing the number of mice analyzed per group, the total number of cholangiocytes quantified per group and the amount of tdTomato+ cholangiocytes in absolute values and percentages. Each row represents a different condition. Opn-Cre, *Opn-CreER-tdTomato* mice; Ck19-Cre, *Ck19-CreER-tdTomato* mice; TAM, Tamoxifen.

**Table 2 cells-08-00380-t002:** Quantification of tdTomato+ hepatocytes.

Treatment Group	Mice (n)		$\frac{\#\;\mathrm{Hnf}-4\alpha^{+}\;{\mathrm{tdTomato}}^{+}}{\#\;\mathrm{Hnf}-4\alpha^{+}}$		% tdTomato^+^ of Hnf-4α^+^	
	Opn-Cre	Ck19-Cre	Opn-Cre	Ck19-Cre	Opn-Cre	Ck19-Cre
uninjured 3 days after TAM	7	7	5/67462	0/61190	0.0074 ± 0.0126	0
uninjured 3 weeks after TAM	8	8	7/77086	0/76513	0.0091 ± 0.0098	0
CCl_4_ 3-day washout	4	5	0/35801	0/44034	0	0
CCl_4_ 3-week washout	3	4	7/30984	0/40768	0.0226 ± 0.0371	0
CCl_4_ no washout ^†^	5	5	81/24192	1/21016	0.3348 ± 0.3588	0.0048 ± 0.0091

† 9 pictures from 1 liver section were counted per mouse, instead of 18 pictures from 2 sections as in the other groups. Table showing the number of mice analyzed per group, the total number of hepatocytes quantified per group and the amount of tdTomato^+^ hepatocytes in absolute values and percentages. Each row represents a different condition. Opn-Cre, *Opn-CreER-tdTomato* mice; Ck19-Cre, *Ck19-CreER-tdTomato* mice; TAM, Tamoxifen.

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
