# Peer review of "Comparison of the Opn-CreER and Ck19-CreER Drivers in Bile Ducts of Normal and Injured Mouse Livers"

_cells, 2019, doi:10.3390/cells8040380_

Round 1
Reviewer 1 Report
To editors and authors:
This is a well-done study directly comparing efficiency and specificity of two commonly used tamoxifen-inducible CreER mouse models for biliary recombination of floxed alleles (Opn-CreER and Ck19-CreER).
The manuscript is well-written, data presentation is clear, and although the results are not entirely surprising, such studies are very important for the field and will likely have impact on the interpretation of previous and future lineage tracing studies and will help other researchers to choose the right mouse model for their studies.
As outline below, I have some minor concerns with data presentation but also highly encourage the authors to present a more balanced discussion that clearly states the limitations of the study which does not support the rather broad overall conclusion made in the paper (OpnCreER: “optimally suited for generation entirely mutant bile ducts, Ck19CreER “favorable for lineage tracing”). Further, several questions should be clarified before publication, which can be mostly done by a more balanced overall conclusion and discussion.
1) The main conclusion made by the authors is that the “OpnCreER driver is optimally suited for generation of entirely mutant bile ducts, while Ck19CreER is favorable to trace the lineage of bile duct cells in the liver”. Further, they conclude that “ectopic recombination in hepatocytes can be avoided by applying a three-day long wash-out period of tamoxifen”. However, while recombination efficacy of the OpnCreER is clearly superior over the CK19CreER strain, such general conclusion cannot be derived from the very specific experimental setup used in this study for several reasons:
The authors use the R26-tdTomato reporter strain ((Madisen et al., 2010), this reference is missing in the reference list!), which is known to be highly sensitive to Cre-mediated recombination over other reporter strains, likely due close distance and easy-to-access loxP sites within the R26 locus (Liu et al., 2013). Accordingly, previous studies utilizing the OpnCreER strain for biliary lineage tracing using other reporter strains (i.e. YFP reporter) find markedly less labeling (about 70-80% vs. almost 100% in the present study). Recombination efficiency may be markedly less in case biliary conditional gene knockouts (requiring recombination of both alleles) shall be produced.
Therefore, the authors must either tone done their rather broad and general conclusion of “perfect recombination efficacy to produce entirely mutant bile ducts” to the specific reporter strain used in this study (recombination of only one “easy-to-access” allele of a R26-tdTomato strain) or, alternatively, demonstrate that the OpnCreER mouse is capable to recombine various other floxed genes with equal efficiency.
Further, the authors conclude that the CK19CreER strain has “near absolute specificity” for the biliary compartment and therefore they state that this strain is favorable for lineage tracing. However, here, several caveats must be considered:
Lineage tracing requires inducible and reversible (here CreERT2) activation of Cre. However, and this is actually one of the important findings in this study, there is leaky active Cre independent of Tamoxifen at basal!. Leakiness especially in knock-in CreER strains is commonly seen (at least to some degree) and is one of the really important findings for the Ck19CreER strain (at least when using the very sensitive tdTom-reporter). The authors state that 50% of “bile ducts” in Ck19CreER and “1/20” bile ducts in OpnCreER mice. From the singular cross section of an interlobular bile duct in Figure 1b one would expect about 20% cholangiocytes with leaky tdTom expression. The exact quantification of this “leakiness” must be provided (absolute number of cholangiocytes labeled at basal given in percentage).
Considering that there may be 10-20% of all cholangiocytes labeled at basal in the CK19CreER strain in the absence of tamoxifen, this may produce false positive lineage labeling of hepatocytes in models where hepatocytes express CK19 (i.e. long term cholestatic models over several months) that cannot be controlled even by infinite washout for tamoxifen. This may be the reason why one recent study identified “biliary-derived” hepatocytes after long term DDC treatment (24 weeks) using the CK19CreER knockin strain (Deng et al., 2018) while previous studies failed to demonstrate biliary-to-hepatocyte neogenesis after even longer DDC-treatment using the Hnf1bCreER strain, i.e. (Jors et al., 2015). In any case, great care must be taken when using the CK19CreER mouse strain, at least when using a sensitive reporter in combination with injury models with expression of biliary proteins.
The authors also state in the abstract that a three-day washout period of tamoxifen avoids “ectopic Cre activation” in injury. This may only hold true for the specific experimental setup in this study and this should be made clear in the text. It is well known from several studies that especially after high or repetitive doses of tamoxifen longer washout times are required, again, depending on Cre-strain, reporter-strain, and, equally important, the type of the specific injury model chosen. Here, the model used (short-term CCl4) is not the model where one would expect dramatic rise in expression of biliary proteins in hepatocytes. As stated above, other models like bile duct ligation or DDC (especially long-term) will lead to stronger Opn and even CK19-expression over time.
2) There is a substantial mess with the data in table 1 and table 2: most percentages (%) do not match with the absolute cell numbers quantified (also often do not match percentages in the text, further, Fig. 3C absolute cell numbers for Ck19-CreER and OpnCreER seem to be mixed up). The authors should go through all these data, make sure absolute cell numbers are correct, and re-calculate the respective percentages!
3) Page 2 line 71: ref 6 used a tdTomato reporter, ref. 7 used a YFP reporter.
4) The authors should provide a figure (best placed in Figure 1: co-stainings tdTomato/CK19 showing entire portal field) where the reader can clearly see that, besides interlobular ducts (as presented in Fig. 1b, which does not show an entire portal field), also biliary cells/ductules of the periportal plexus are readily identified with the Troma III CK19 antibody (otherwise there would be substantial over-estimation of labelling efficacy of the biliary compartment for both Cre strains).
Deng, X., Zhang, X., Li, W., Feng, R.X., Li, L., Yi, G.R., Zhang, X.N., Yin, C., Yu, H.Y., Zhang, J.P., et al. (2018). Chronic Liver Injury Induces Conversion of Biliary Epithelial Cells into Hepatocytes. Cell Stem Cell 23, 114-122 e113.
Jors, S., Jeliazkova, P., Ringelhan, M., Thalhammer, J., Durl, S., Ferrer, J., Sander, M., Heikenwalder, M., Schmid, R.M., Siveke, J.T., et al. (2015). Lineage fate of ductular reactions in liver injury and carcinogenesis. J Clin Invest 125, 2445-2457.
Liu, J., Willet, S.G., Bankaitis, E.D., Xu, Y., Wright, C.V., and Gu, G. (2013). Non-parallel recombination limits Cre-LoxP-based reporters as precise indicators of conditional genetic manipulation. Genesis 51, 436-442.
Madisen, L., Zwingman, T.A., Sunkin, S.M., Oh, S.W., Zariwala, H.A., Gu, H., Ng, L.L., Palmiter, R.D., Hawrylycz, M.J., Jones, A.R., et al. (2010). A robust and high-throughput Cre reporting and characterization system for the whole mouse brain. Nat Neurosci 13, 133-140.
Author Response
Point 1: The main conclusion made by the authors is that the “OpnCreER driver is optimally suited for generation of entirely mutant bile ducts, while Ck19CreER is favorable to trace the lineage of bile duct cells in the liver”. Further, they conclude that “ectopic recombination in hepatocytes can be avoided by applying a three-day long wash-out period of tamoxifen”. However, while recombination efficacy of the OpnCreER is clearly superior over the CK19CreER strain, such general conclusion cannot be derived from the very specific experimental setup used in this study for several reasons:
The authors use the R26-tdTomato reporter strain ((Madisen et al., 2010), this reference is missing in the reference list!), which is known to be highly sensitive to Cre-mediated recombination over other reporter strains, likely due close distance and easy-to-access loxP sites within the R26 locus (Liu et al., 2013). Accordingly, previous studies utilizing the OpnCreER strain for biliary lineage tracing using other reporter strains (i.e. YFP reporter) find markedly less labeling (about 70-80% vs. almost 100% in the present study). Recombination efficiency may be markedly less in case biliary conditional gene knockouts (requiring recombination of both alleles) shall be produced.
Therefore, the authors must either tone done their rather broad and general conclusion of “perfect recombination efficacy to produce entirely mutant bile ducts” to the specific reporter strain used in this study (recombination of only one “easy-to-access” allele of a R26-tdTomato strain) or, alternatively, demonstrate that the OpnCreER mouse is capable to recombine various other floxed genes with equal efficiency.
Response 1: The reference for the reporter strain was added, results from previous studies with the OpnCreER strain were mentioned in the discussion and the conclusions toned down as requested by the reviewer. The discussion now reads:
“Notably, other studies reported lower levels of recombination with the Opn-iCreERT2 driver, probably because lower amounts of tamoxifen were injected and a less sensitive reporter strain was used (Rosa26-loxP-STOP-loxP-YFP).” (Line 287 – 290)
“Despite the fact that Opn-iCreERT2 is a stronger recombination driver than Ck19-CreERT, one should keep in mind that efficiency of creating knockout cells may be lower than 100% in knockout studies. Recombination of one allele is sufficient to produce a positive signal from the R26-tdTomato reporter yet generation of homozygous mutant cells may require recombination of both alleles. In addition, the sensitivity to Cre recombination for a floxed allele may be lower than for the tdTomato reporter line which is known to be very sensitive [20]. Determination of the recombination efficiency is therefore important to interpret the results of knockout experiments.” (Line 293 – 299)
The conclusion now reads: “It can be concluded that the Opn-iCreERT2 driver is best suited for the generation of mutant bile ducts, while the Ck19-CreERT driver has near absolute specificity for bile duct cells and is therefore favorable for lineage tracing experiments. Importantly however, it has to be considered in this model that Cre recombination is present in the absence of tamoxifen.” (Line 340 - 343)
Further, the authors conclude that the CK19CreER strain has “near absolute specificity” for the biliary compartment and therefore they state that this strain is favorable for lineage tracing. However, here, several caveats must be considered:
Lineage tracing requires inducible and reversible (here CreERT2) activation of Cre. However, and this is actually one of the important findings in this study, there is leaky active Cre independent of Tamoxifen at basal!. Leakiness especially in knock-in CreER strains is commonly seen (at least to some degree) and is one of the really important findings for the Ck19CreER strain (at least when using the very sensitive tdTom-reporter). The authors state that 50% of “bile ducts” in Ck19CreER and “1/20” bile ducts in OpnCreER mice. From the singular cross section of an interlobular bile duct in Figure 1b one would expect about 20% cholangiocytes with leaky tdTom expression. The exact quantification of this “leakiness” must be provided (absolute number of cholangiocytes labeled at basal given in percentage).
Considering that there may be 10-20% of all cholangiocytes labeled at basal in the CK19CreER strain in the absence of tamoxifen, this may produce false positive lineage labeling of hepatocytes in models where hepatocytes express CK19 (i.e. long term cholestatic models over several months) that cannot be controlled even by infinite washout for tamoxifen. This may be the reason why one recent study identified “biliary-derived” hepatocytes after long term DDC treatment (24 weeks) using the CK19CreER knockin strain (Deng et al., 2018) while previous studies failed to demonstrate biliary-to-hepatocyte neogenesis after even longer DDC-treatment using the Hnf1bCreER strain, i.e. (Jors et al., 2015). In any case, great care must be taken when using the CK19CreER mouse strain, at least when using a sensitive reporter in combination with injury models with expression of biliary proteins.
The quantification data of tdTomato leakiness in absence of tamoxifen were expanded and added in the results section. The results now read: “After corn oil injection (vehicle), Opn-CreER-tdTomato mice showed tdTomato expression in 2% of cholangiocytes (9/388). The 9 tdTomato+ cholangiocytes were found in 7 different bile ducts and usually appeared as single cells. In Ck19-CreER-tdTomato corn oil injected mice, 13% of cholangiocytes were tdTomato+ (54/418) (Fig.1b). The 54 tdTomato+ cells were found in 8 different bile ducts and usually appeared as large patches of cells, presumably clones, in large bile ducts. This amounted to 1.80 independent recombination events (clones) per 100 cholangiocytes for Opn-CreERT-tdTomato mice and 1.91 for Ck19-CreERT-tdTomato mice observed at 8 weeks of age.” (Line 173 – 184)
The discussion now reads: “Our study showed Cre activity in the absence of tamoxifen. Recombination occurred in 13% of cholangiocytes in the Ck19-CreERT-tdTomato line and in 2% in the Opn-iCre-ERT2-tdTomato line. Despite its high specificity, the CK19-CreERT might therefore be unsuited for lineage tracing experiments since temporal control over Cre recombination is partially lost. These baseline levels of Cre recombination have to be considered when performing experiments.” (Line 300 - 304)
The authors also state in the abstract that a three-day washout period of tamoxifen avoids “ectopic Cre activation” in injury. This may only hold true for the specific experimental setup in this study and this should be made clear in the text. It is well known from several studies that especially after high or repetitive doses of tamoxifen longer washout times are required, again, depending on Cre-strain, reporter-strain, and, equally important, the type of the specific injury model chosen. Here, the model used (short-term CCl4) is not the model where one would expect dramatic rise in expression of biliary proteins in hepatocytes. As stated above, other models like bile duct ligation or DDC (especially long-term) will lead to stronger Opn and even CK19-expression over time.
The length of washout period after tamoxifen is dependent on the experimental setup and this is mentioned in the discussion: “The length of such a washout period will however depend on the tamoxifen regimen applied and on the injury model, and should therefore be re-examined when using a different experimental set-up.” (Line 323 – 325)
Point 2: There is a substantial mess with the data in table 1 and table 2: most percentages (%) do not match with the absolute cell numbers quantified (also often do not match percentages in the text, further, Fig. 3C absolute cell numbers for Ck19-CreER and OpnCreER seem to be mixed up). The authors should go through all these data, make sure absolute cell numbers are correct, and re-calculate the respective percentages!
Response 2: We want to thank the reviewer to point out mistake. This comment was addressed by correcting the numbers in the tables and text. All percentages now match the absolute numbers.
Point 3: Page 2 line 71: ref 6 used a tdTomato reporter, ref. 7 used a YFP reporter.
Response 3: The use of YFP reporter was specified in the text.
Point 4: The authors should provide a figure (best placed in Figure 1: co-stainings tdTomato/CK19 showing entire portal field) where the reader can clearly see that, besides interlobular ducts (as presented in Fig. 1b, which does not show an entire portal field), also biliary cells/ductules of the periportal plexus are readily identified with the Troma III CK19 antibody (otherwise there would be substantial over-estimation of labelling efficacy of the biliary compartment for both Cre strains).
Response 4: An extra supplemental figure was added illustrating an entire portal field and the distribution of Ck19 staining in all biliary cells, as well as tdTomato expression after tamoxifen injections (Fig.S1).
Reviewer 2 Report
In this manuscript, the authors characterized the specificity and efficiency of two cholangiocyte-specific Cre drivers, CK19-CreER and Opn-CreER. They showed that the CK19-Cre driver is highly specific but has relatively low efficiency. The Opn-Cre driver has very high efficiency of Cre recombination, but can be activated in hepatocytes upon acute CCl4-induced injury. Given the broad use of these two Cre-drivers in labeling cholangiocyte, the current study is very informative for the studies of cholangiocyte biology in normal and injured liver.
Main comments:
1. In the current study, tamoxifen was administered to young adult mice. It will be interesting to characterize the specificity and efficiency of Cre recombination in healthy and injured livers in neonates.
2. The pathohistology of CCl4-induced acute liver injury is not shown.
3. The authors showed that acute CCl4 treatment induced ectopic activation of Opn-Cre in hepatocytes. Where are these Opn-positive hepatocytes located? Are they restricted to a specific zone?
Author Response
Point 1: In the current study, tamoxifen was administered to young adult mice. It will be interesting to characterize the specificity and efficiency of Cre recombination in healthy and injured livers in neonates.
Response 1: In our report we specifically addressed adult mouse lines since we aim to study liver regeneration in adults, rather than investigate compensatory proliferation in neonates.
Point 2: The pathohistology of CCl4-induced acute liver injury is not shown.
Response 2: We addressed this comment by adding a supplemental H&E picture (Figure S4), showing necrotic hepatocytes in a liver section 48h after CCl4 treatment.
Point 3: The authors showed that acute CCl4 treatment induced ectopic activation of Opn-Cre in hepatocytes. Where are these Opn-positive hepatocytes located? Are they restricted to a specific zone?
Response 3: Hepatocytes inducing activation of Opn-Cre were found in all zones of the liver and are equally distributed. The discussion reads: “These tdTomato+ hepatocytes usually appeared as single cells and their location was not restricted to a specific zone.” (Line 228-229)
Reviewer 3 Report
The article by Bram Lesaffer et al. is an original contribution describing and comparing Opn-CreER and Ck19-CreER driver in bile ducts of normal and injured mouse livers. Introduction and discussion are sufficient. The methods are well designed and executed. Authors described all the experiments in a very detailed way assuring the possibility of repeating the experiments. Figures and tables are adequate.
I noticed solely few minor spelling mistakes which should be corrected by authors.
Author Response
Point 1: I noticed solely few minor spelling mistakes which should be corrected by authors.
Response 1: All spelling errors were corrected.
Round 2
Reviewer 1 Report
The authors have addressed all points raised.
Reviewer 2 Report
The authors properly addressed my comments.